# Attend and Predict: Understanding Gene Regulation by Selective Attention on Chromatin

**Ritambhara Singh, Jack Lanchantin, Arshdeep Sekhon, Yanjun Qi**
Department of Computer Science
University of Virginia
yanjun@virginia.edu

## Abstract

The past decade has seen a revolution in genomic technologies that enabled a flood of genome-wide profiling of chromatin marks. Recent literature tried to understand gene regulation by predicting gene expression from large-scale chromatin measurements. Two fundamental challenges exist for such learning tasks: (1) genome-wide chromatin signals are spatially structured, high-dimensional and highly modular; and (2) the core aim is to understand what the relevant factors are and how they work together. Previous studies either failed to model complex dependencies among input signals or relied on separate feature analysis to explain the decisions. This paper presents an attention-based deep learning approach, AttentiveChrome, that uses a unified architecture to model and to interpret dependencies among chromatin factors for controlling gene regulation. AttentiveChrome uses a hierarchy of multiple Long Short-Term Memory (LSTM) modules to encode the input signals and to model how various chromatin marks cooperate automatically. AttentiveChrome trains two levels of attention jointly with the target prediction, enabling it to attend differentially to relevant marks and to locate important positions per mark. We evaluate the model across 56 different cell types (tasks) in humans. Not only is the proposed architecture more accurate, but its attention scores provide a better interpretation than state-of-the-art feature visualization methods such as saliency maps.[1]

## 1   Introduction

Gene regulation is the process of how the cell controls which genes are turned "on" (expressed) or "off" (not-expressed) in its genome. The human body contains hundreds of different cell types, from liver cells to blood cells to neurons. Although these cells include the same set of DNA information, their functions are different [2]. The regulation of different genes controls the destiny and function of each cell. In addition to DNA sequence information, many factors, especially those in its environment (i.e., chromatin), can affect which genes the cell expresses. This paper proposes an attention-based deep learning architecture to learn from data how different chromatin factors influence gene expression in a cell. Such understanding of gene regulation can enable new insights into principles of life, the study of diseases, and drug development.

"Chromatin" denotes DNA and its organizing proteins [3]. A cell uses specialized proteins to organize DNA in a condensed structure. These proteins include histones, which form "bead"-like structures that DNA wraps around, in turn organizing and compressing the DNA. An important aspect of histone proteins is that they are prone to chemical modifications that can change the spatial arrangement of

DNA. These spatial re-arrangements result in certain DNA regions becoming accessible or restricted and therefore affecting expressions of genes in the neighborhood region. Researchers have established the "Histone Code Hypothesis" that explores the role of histone modifications in controlling gene regulation. Unlike genetic mutations, chromatin changes such as histone modifications are potentially reversible ([5]). This crucial difference makes the understanding of how chromatin factors determine gene regulation even more impactful because this knowledge can help developing drugs targeting genetic diseases.

At the whole genome level, researchers are trying to chart the locations and intensities of all the chemical modifications, referred to as marks, over the chromatin [4]. Recent advances in next-generation sequencing have allowed biologists to profile a significant amount of gene expression and chromatin patterns as signals (or read counts) across many cell types covering the full human genome. These datasets have been made available through large-scale repositories, the latest being the Roadmap Epigenome Project (REMC, publicly available) ([18]). REMC recently released 2,804 genome-wide datasets, among which 166 datasets are gene expression reads (RNA-Seq datasets) and the rest are signal reads of various chromatin marks across 100 different "normal" human cells/tissues [18].

The fundamental aim of processing and understanding this repository of "big" data is to understand gene regulation. For each cell type, we want to know which chromatin marks are the most important and how they work together in controlling gene expression. However, previous machine learning studies on this task either failed to model spatial dependencies among marks or required additional feature analysis to explain the predictions (Section 4). Computational tools should consider two important properties when modeling such data.

- First, signal reads for each mark are spatially structured and high-dimensional. For instance, to quantify the influence of a histone modification mark, learning methods typically need to use as input features all of the signals covering a DNA region of length $10,000$ base pair (bp) [5] centered at the transcription start site (TSS) of each gene. These signals are sequentially ordered along the genome direction. To develop "epigenetic" drugs, it is important to recognize how a chromatin mark's effect on regulation varies over different genomic locations.
- Second, various types of marks exist in human chromatin that can influence gene regulation. For example, each of the five standard histone proteins can be simultaneously modified at multiple different sites with various kinds of chemical modifications, resulting in a large number of different histone modification marks. For each mark, we build a feature vector representing its signals surrounding a gene's TSS position. When modeling genome-wide signal reads from multiple marks, learning algorithms should take into account the modular nature of such feature inputs, where each mark functions as a module. We want to understand how the interactions among these modules influence the prediction (gene expression).

In this paper we propose an attention-based deep learning model, AttentiveChrome, that learns to predict the expression of a gene from an input of histone modification signals covering the gene's neighboring DNA region. By using a hierarchy of multiple LSTM modules, AttentiveChrome can discover interactions among signals of each chromatin mark, and simultaneously learn complex dependencies among different marks. Two levels of "soft" attention mechanisms are trained, (1) to attend to the most relevant regions of a chromatin mark, and (2) to recognize and attend to the important marks. Through predicting and attending in one unified architecture, AttentiveChrome allows users to understand how chromatin marks control gene regulation in a cell. In summary, this work makes the following contributions:

- AttentiveChrome provides more accurate predictions than state-of-the-art baselines. Using datasets from REMC, we evaluate AttentiveChrome on 56 different cell types (tasks).
- We validate and compare interpretation scores using correlation to a new mark signal from REMC (not used in modeling). AttentiveChrome's attention scores provide a better interpretation than state-of-the-art methods for visualizing deep learning models.
- AttentiveChrome can model highly modular inputs where each module is highly structured. AttentiveChrome can explain its decisions by providing "what" and "where" the model has focused

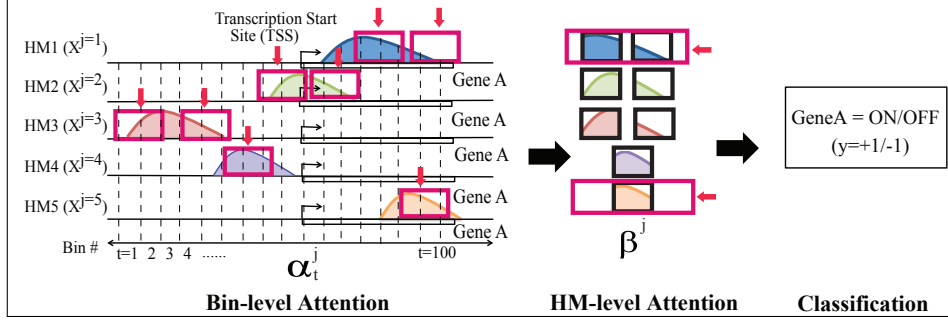

Figure 1: Overview of the proposed AttentiveChrome framework. It includes 5 important parts: (1) Bin-level LSTM encoder for each HM mark; (2) Bin-level $\alpha$-Attention across all bin positions of each HM mark; (3) HM-level LSTM encoder encoding all HM marks; (4) HM-level $\beta$-Attention among all HM marks; (5) the final classification.

on. This flexibility and interpretability make this model an ideal approach for many real-world applications.

- To the authors' best knowledge, AttentiveChrome is the first attention-based deep learning method for modeling data from molecular biology.

In the following sections, we denote vectors with bold font and matrices using capital letters. To simplify notation, we use "HM" as a short form for the term "histone modification".

## 2   Background: Long Short-Term Memory (LSTM) Networks

Recurrent neural networks (RNNs) have been widely used in applications such as natural language processing due to their abilities to model sequential dependencies. Given an input matrix $\mathbf{X}$ of size $n_{in} \times T$, an RNN produces a matrix $\mathbf{H}$ of size $d \times T$, where $n_{in}$ is the input feature size, $T$ is the length of input feature , and $d$ is the RNN embedding size. At each timestep $t \in \{1, \cdots, T\}$, an RNN takes an input column vector $\mathbf{x}_t \in \mathbb{R}^{n_{in}}$ and the previous hidden state vector $\mathbf{h_{t-1}} \in \mathbb{R}^d$ and produces the next hidden state $\mathbf{h_t}$ by applying the following recursive operation:

$$\mathbf{h}_t = \sigma(\mathbf{W}\mathbf{x}_t + \mathbf{U}\mathbf{h}_{t-1} + \mathbf{b}) = \overrightarrow{LSTM}(\mathbf{x}_t), \qquad (1)$$

where $\mathbf{W}, \mathbf{U}, \mathbf{b}$ are the trainable parameters of the model, and $\sigma$ is an element-wise nonlinearity function. Due to the recursive nature, RNNs can capture the complete set of dependencies among all timesteps being modeled, like all spatial positions in a sequential sample. To handle the "vanishing gradient" issue of training basic RNNs, Hochreiter *et al.* [13] proposed an RNN variant called the Long Short-term Memory (LSTM) network.

An LSTM layer has an input-to-state component and a recurrent state-to-state component like that in Eq. (1). Additionally, it has gating functions that control when information is written to, read from, and forgotten. Though the LSTM formulation results in a complex form of Eq. (1) (see Supplementary), when given input vector $\mathbf{x}_t$ and the state $\mathbf{h}_{t-1}$ from previous time step $t-1$, an LSTM module also produces a new state $\mathbf{h}_t$. The embedding vector $\mathbf{h}_t$ encodes the learned representation summarizing feature dependencies from the time step 0 to the time step $t$. For our task, we call each bin position on the genome coordinate a "time step".

## 3   AttentiveChrome: A Deep Model for Joint Classification and Visualization

**Input and output formulation for the task:** We use the same feature inputs and outputs as done previously in DeepChrome ([29]). Following Cheng *et al.* [7], the gene expression prediction is formulated as a binary classification task whose output represents if the gene expression of a gene is high(+1) or low(-1). As shown in Figure 1, the input feature of a sample (a particular gene) is denoted as a matrix $\mathbf{X}$ of size $M \times T$. Here $M$ denotes the number of HM marks we consider in the input. $T$ is the total number of bin positions we take into account from the neighboring region of a gene's TSS site on the genome. We refer to this region as the 'gene region' in the rest of the paper. $\mathbf{x}^j$ denotes the $j$-th row vector of $\mathbf{X}$ whose elements are sequentially structured (signals from the $j$-th HM mark) $j \in \{1, ..., M\}$. $x_t^j$ in matrix $\mathbf{X}$ represents the signal from the $t$-th bin of the $j$-th HM

mark. $t \in \{1, ..., T\}$. We assume our training set $D$ contains $N_{tr}$ labeled pairs. We denote the $n$-th pair as $(\mathbf{X}^{(n)}, y^{(n)})$, $\mathbf{X}^{(n)}$ is a matrix of size $M \times T$ and $y^{(n)} \in \{-1, +1\}$, where $n \in \{1, ..., N_{tr}\}$.

**An end-to-end deep architecture for predicting and attending jointly:** AttentiveChrome learns to predict the expression of a gene from an input of HM signals covering its gene region. First, the signals of each HM mark are fed into a separate LSTM network to encode the spatial dependencies among its bin signals, and then another LSTM is used to model how multiple factors work together for predicting gene expression. Two levels of "soft" attention mechanisms are trained and dynamically predicted for each gene: (1) to attend to the most relevant positions of an HM mark, and (2) then to recognize and attend to the relevant marks. In summary, AttentiveChrome consists of five main modules (see Supplementary Figure S:2): (1) Bin-level LSTM encoder for each HM mark; (2) Bin-level Attention on each HM mark; (3) HM-level LSTM encoder encoding all HM marks; (4) HM-level Attention over all the HM marks; (5) the final classification module. We describe the details of each component as follows:

**Bin-Level Encoder Using LSTMs:** For a gene of interest, the $j$-th row vector, $\mathbf{x}^j$, from $\mathbf{X}$ includes a total of $T$ elements that are sequentially ordered along the genome coordinate. Considering the sequential nature of such signal reads, we treat each element (essentially a bin position) as a 'time step' and use a bidirectional LSTM to model the complete dependencies among elements in $\mathbf{x}^j$. A bidirectional LSTM contains two LSTMs, one in each direction (see Supplementary Figure S:2 (c)). It includes a forward $\overrightarrow{LSTM}^j$ that models $\mathbf{x}^j$ from $x_1^j$ to $x_T^j$ and a backward $\overleftarrow{LSTM}^j$ that models from $x_T^j$ to $x_1^j$. For each position $t$, the two LSTMs output $\overrightarrow{\mathbf{h}_t^j}$ and $\overleftarrow{\mathbf{h}_t^j}$, each of size $d$. $\overrightarrow{\mathbf{h}_t^j} = \overrightarrow{LSTM}^j(x_t^j)$ and $\overleftarrow{\mathbf{h}_t^j} = \overleftarrow{LSTM}^j(x_t^j)$. The final embedding vector at the $t$-th position is the concatenation $\mathbf{h}_t^j = [\overrightarrow{\mathbf{h}_t^j}, \overleftarrow{\mathbf{h}_t^j}]$.

By coupling these LSTM-based HM encoders with the final classification, they can learn to embed each HM mark by extracting the dependencies among bins that are essential for the prediction task.

**Bin-Level Attention, $\alpha$-attention:** Although the LSTM can encode dependencies among the bins, it is difficult to determine which bins are *most important* for prediction from the LSTM. To automatically and adaptively highlight the most relevant bins for each sample, we use "soft" attention to learn the importance weights of bins. This means when representing $j$-th HM mark, AttentiveChrome follows a basic concept that not all bins contribute equally to the encoding of the entire $j$-th HM mark. The attention mechanism can help locate and recognize those bins that are important for the current gene sample of interest from $j$-th HM mark and can aggregate those important bins to form an embedding vector. This extraction is implemented through learning a weight vector $\boldsymbol{\alpha}^j$ of size $T$ for the $j$-th HM mark. For $t \in \{1, ..., T\}$, $\alpha_t^j$ represents the importance of the $t$-th bin in the $j$-th HM. It is computed as: $\alpha_t^j = \frac{exp(\mathbf{W}_b\mathbf{h}_t^j)}{\sum_{i=1}^{T} exp(\mathbf{W}_b\mathbf{h}_i^j)}$.

$\alpha_t^j$ is a scalar and is computed by considering all bins' embedding vectors $\{\mathbf{h}_1^j, \cdots, \mathbf{h}_T^j\}$. The context parameter $\mathbf{W}_b$ is randomly initialized and jointly learned with the other model parameters during training. Our intuition is that through $\mathbf{W}_b$ the model will automatically learn the context of the task (e.g., type of a cell) as well as the positional relevance to the context simultaneously. Once we have the importance weight of each bin position, we can represent the entire $j$-th HM mark as a weighted sum of all its bin embeddings: $\mathbf{m}^j = \sum_{t=1}^{T} \alpha_t^j \times \mathbf{h}_t^j$. Essentially the attention weights $\alpha_t^j$ tell us the relative importance of the $t$-th bin in the representation $\mathbf{m}^j$ for the current input $\mathbf{X}$ (both $\mathbf{h}_t^j$ and $\alpha_t^j$ depend on $\mathbf{X}$).

**HM-Level Encoder Using Another LSTM:** We aim to capture the dependencies among HMs as some HMs are known to work together to repress or activate gene expression [6]. Therefore, next we model the joint dependencies among multiple HM marks (essentially, learn to represent a set). Even though there exists no clear order among HMs, we assume an imagined sequence as $\{HM_1, HM_2, HM_3, ..., HM_M\}$ [6]. We implement another bi-directional LSTM encoder, this time on the imagined sequence of HMs using the representations $\mathbf{m}^j$ of the $j$-th HMs as LSTM inputs (Supplementary Figure S:2 (e)). Setting the embedding size as $d'$, this set-based encoder, we denote as $LSTM_s$, encodes the $j$-th HM as: $\mathbf{s}^j = [\overrightarrow{LSTM_s}(\mathbf{m}^j), \overleftarrow{LSTM_s}(\mathbf{m}^j)]$. Differently from $\mathbf{m}^j$, $\mathbf{s}^j$ encodes the dependencies between the $j$-th HM and other HM marks.

Table 1: Comparison of previous studies for the task of quantifying gene expression using histone modification marks (adapted from [29]). AttentiveChrome is the only model that exhibits all 8 desirable properties.

| Computational Study | Unified | Non-linear | Bin-Info | Representation Learning | | Prediction | Feature Inter. | Interpretable |
|---|---|---|---|---|---|---|---|---|
| | | | | Neighbor Bins | Whole Region | | | |
| Linear Regression ([14]) | ✗ | ✗ | ✗ | ✗ | ✓ | ✓ | ✗ | ✓ |
| Support Vector Machine ([7]) | ✗ | ✓ | Bin-specific | ✗ | ✓ | ✓ | ✓ | ✗ |
| Random Forest ([10]) | ✗ | ✓ | Best-bin | ✗ | ✓ | ✓ | ✗ | ✗ |
| Rule Learning ([12]) | ✗ | ✓ | ✗ | ✗ | ✓ | ✗ | ✓ | ✓ |
| DeepChrome-CNN [29] | ✓ | ✓ | Automatic | ✓ | ✓ | ✓ | ✓ | ✗ |
| **AttentiveChrome** | ✓ | ✓ | Automatic | ✓ | ✓ | ✓ | ✓ | ✓ |

**HM-Level Attention, $\beta$-attention:** Now we want to focus on the important HM markers for classifying a gene's expression as high or low. We do this by learning a second level of attention among HMs. Similar to learning $\alpha_t^j$, we learn another set of weights $\beta^j$ for $j \in \{1, \cdots, M\}$ representing the importance of HM$^j$. $\beta^i$ is calculated as: $\beta^j = \frac{exp(\mathbf{W}_s \mathbf{s}^j)}{\sum_{i=1}^{M} exp(\mathbf{W}_s \mathbf{s}^i)}$. The *HM-level* context parameter $\mathbf{W}_s$ learns the context of the task and learns how HMs are relevant to that context. $\mathbf{W}_s$ is randomly initialized and jointly trained. We encode the entire "gene region" into a hidden representation $\mathbf{v}$ as a weighted sum of embeddings from all HM marks: $\mathbf{v} = \sum_{j=1}^{M} \beta^j \mathbf{s}^j$. We can interpret the learned attention weight $\beta^i$ as the relative importance of HM$^i$ when blending all HM marks to represent the entire gene region for the current gene sample $\mathbf{X}$.

**Training AttentiveChrome End-to-End:** The vector $\mathbf{v}$ summarizes the information of all HMs for a gene sample. We feed it to a simple classification module $f$ (Supplementary Figure S:2(f)) that computes the probability of the current gene being expressed high or low: $f(\mathbf{v}) = \text{softmax}(\mathbf{W}_c \mathbf{v} + b_c)$. $\mathbf{W}_c$ and $b_c$ are learnable parameters. Since the entire model, including the attention mechanisms, is differentiable, learning end-to-end is trivial by using backpropagation [21]. All parameters are learned together to minimize a negative log-likelihood loss function that captures the difference between true labels $y$ and predicted scores from $f(.)$.

# 4 Connecting to Previous Studies

In recent years, there has been an explosion of deep learning models that have led to groundbreaking performance in many fields such as computer vision [17], natural language processing [30], and computational biology [1, 27, 38, 16, 19, 29].

**Attention-based deep models:** The idea of attention in deep learning arises from the properties of the human visual system. When perceiving a scene, the human vision gives more importance to some areas over others [9]. This adaptation of "attention" allows deep learning models to focus selectively on only the important features. Deep neural networks augmented with attention mechanisms have obtained great success on multiple research topics such as machine translation [4], object recognition [2, 26], image caption generation [33], question answering [30], text document classification [34], video description generation[35], visual question answering -[32], or solving discrete optimization [31]. Attention brings in two benefits: (1) By selectively focusing on parts of the input during prediction the attention mechanisms can reduce the amount of computation and the number of parameters associated with deep learning model [2, 26]. (2) Attention-based modeling allows for learning salient features dynamically as needed [34], which can help improve accuracy.

Different attention mechanisms have been proposed in the literature, including 'soft' attention [4], 'hard' attention [33, 24], or 'location-aware' [8]. Soft attention [4] calculates a 'soft' weighting scheme over all the component feature vectors of input. These weights are then used to compute a weighted combination of the candidate feature vectors. The magnitude of an attention weight correlates highly with the degree of significance of the corresponding component feature vector to the prediction. Inspired by [34], AttentiveChrome uses two levels of soft attention for predicting gene expression from HM marks.

**Visualizing and understanding deep models:** Although deep learning models have proven to be very accurate, they have widely been viewed as "black boxes". Researchers have attempted to develop separate visualization techniques that explain a deep classifier's decisions. Most prior studies have focused on understanding convolutional neural networks (CNN) for image classifications, including techniques such as "deconvolution" [36], "saliency maps" [3, 28] and "class optimization" based

visualisation [28]. The "deconvolution' approach [36] maps hidden layer representations back to the input space for a specific example, showing those features of an image that are important for classification. "Saliency maps" [28] use a first-order Taylor expansion to linearly approximate the deep network and seek most relevant input features. The "class optimization" based visualization [28] tries to find the best example (through optimization) that maximizes the probability of the class of interest. Recent studies [15, 22] explored the interpretability of recurrent neural networks (RNN) for text-based tasks. Moreover, since attention in models allows for automatically extracting salient features, attention-coupled neural networks impart a degree of interpretability. By visualizing what the model attends to in [34], attention can help gauge the predictive importance of a feature and hence interpret the output of a deep neural network.

**Deep learning in bioinformatics:** Deep learning is steadily gaining popularity in the bioinformatics community. This trend is credited to its ability to extract meaningful representations from large datasets. For instance, multiple recent studies have successfully used deep learning for modeling protein sequences [23, 37] and DNA sequences [1, 20], predicting gene expressions [29], as well as understanding the effects of non-coding variants [38, 27].

**Previous machine learning models for predicting gene expression from histone modification marks:** Multiple machine learning methods have been proposed to predict gene expression from histone modification data (surveyed by Dong *et al.* [11]) including linear regression [14], support vector machines [7], random forests [10], rule-based learning [12] and CNNs [29]. These studies designed different feature selection strategies to accommodate a large amount of histone modification signals as input. The strategies vary from using signal averaging across all relevant positions, to a 'best position' strategy that selected the input signals at the position with the highest correlation to the target gene expression and automatically learning combinatorial interactions among histone modification marks using CNN (called DeepChrome [29]). DeepChrome outperformed all previous methods (see Supplementary) on this task and used a class optimization-based technique for visualizing the learned model. However, this class-level visualization lacks the necessary granularity to understand the signals from multiple chromatin marks at the individual gene level.

Table 1 compares previous learning studies on the same task with AttentiveChrome across seven desirable model properties. The columns indicate properties (1) whether the study has a unified end-to-end architecture or not, (2) if it captures non-linearity among features, (3) how has the bin information been incorporated, (4) if representation of features is modeled on local and (5) global scales, (6) whether gene expression prediction is provided, (7) if combinatorial interactions among histone modifications are modeled, and finally (8) if the model is interpretable. AttentiveChrome is the only model that exhibits all seven properties. Additionally, Section 5 compares the attention weights from AttentiveChrome with the visualization from "saliency map" and "class optimization." Using the correlation to one additional HM mark from REMC, we show that AttentiveChrome provides better interpretation and validation.

## 5 Experiments and Results

**Dataset:** Following DeepChrome [29], we downloaded gene expression levels and signal data of five core HM marks for 56 different cell types archived by the REMC database [18]. Each dataset contains information about both the location and the signal intensity for a mark measured across the whole genome. The selected five core HM marks have been uniformly profiled across all 56 cell types in the REMC study [18]. These five HM marks include (we rename these HMs in our analysis for readability): H3K27me3 as $H_{reprA}$, H3K36me3 as $H_{struct}$, H3K4me1 as $H_{enhc}$, H3K4me3 as $H_{prom}$, and H3K9me3 as $H_{reprB}$. HMs $H_{reprA}$ and $H_{reprB}$ are known to repress the gene expression, $H_{prom}$ activates gene expression, $H_{struct}$ is found over the gene body, and $H_{enhc}$ sometimes helps in activating gene expression.

**Details of the Dataset:** We divided the $10,000$ base pair DNA region $(+/-5000$ bp) around the transcription start site (TSS) of each gene into bins, with each bin containing 100 continuous bp). For each gene in a specific celltype, the feature generation process generated a $5 \times 100$ matrix, $\mathbf{X}$, where columns represent $T(=100)$ different bins and rows represent $M(=5)$ HMs. For each cell type, the gene expression has been quantified for all annotated genes in the human genome and has been normalized. As previously mentioned, we formulated the task of gene expression prediction as a binary classification task. Following [7], we used the median gene expression across all genes for a particular cell type as the threshold to discretize expression values. For each cell type, we divided

Table 2: AUC score performances for different variations of AttentiveChrome and baselines

| | Baselines | | | AttentiveChrome Variations | | | | |
|---|---|---|---|---|---|---|---|---|
| Model | DeepChrome (CNN) [29] | LSTM | CNN-Attn | CNN-$\alpha, \beta$ | LSTM-Attn | LSTM-$\alpha$ | LSTM-$\alpha, \beta$ |
| Mean | 0.8008 | 0.8052 | 0.7622 | 0.7936 | 0.8100 | **0.8133** | 0.8115 |
| Median | 0.8009 | 0.8036 | 0.7617 | 0.7914 | 0.8118 | **0.8143** | 0.8123 |
| Max | **0.9225** | 0.9185 | 0.8707 | 0.9059 | 0.9155 | 0.9218 | 0.9177 |
| Min | 0.6854 | 0.7073 | 0.6469 | 0.7001 | **0.7237** | 0.7250 | 0.7215 |
| Improvement over DeepChrome [29] (out of 56 cell types) | 36 | 0 | 16 | 49 | **50** | 49 |

our set of 19,802 gene samples into three separate, but equal-size folds for training (6601 genes), validation (6601 genes), and testing (6600 genes) respectively.

**Model Variations and Two Baselines:** In Section 3, we introduced three main components of AttentiveChrome to handle the task of predicting gene expression from HM marks: LSTMs, attention mechanisms, and hierarchical attention. To investigate the performance of these components, our experiments compare multiple AttentiveChrome model variations plus two standard baselines.

- DeepChrome [29]: The temporal (1-D) CNN model used by Singh *et al.* [29] for the same classification task. This study did not consider the modular property of HM marks.
- LSTM: We directly apply an LSTM on the input matrix $\mathbf{X}$ without adding any attention. This setup does not consider the modular property of each HM mark, that is, we treat the signals of all HMs at $t$-th bin position as the $t$-th input to LSTM.
- LSTM-Attn: We add one attention layer on the baseline LSTM model over input $\mathbf{X}$. This setup does not consider the modular property of HM marks.
- CNN-Attn: We apply one attention layer over the CNN model from DeepChrome [29], after removing the max-pooling layer to allow bin-level attention for each bin. This setup does not consider the modular property of HM marks.
- LSTM-$\alpha, \beta$: As introduced in Section 3, this model uses one LSTM per HM mark and add one $\alpha$-attention per mark. Then it uses another level of LSTM and $\beta$-attention to combine HMs.
- CNN-$\alpha, \beta$: This considers the modular property among HM marks. We apply one CNN per HM mark and add one $\alpha$-attention per mark. Then it uses another level of CNN and $\beta$-attention to combine HMs.
- LSTM-$\alpha$: This considers the modular property of HM marks. We apply one LSTM per HM mark and add one $\alpha$-attention per mark. Then, the embedding of HM marks is concatenated as one long vector and then fed to a 2-layer fully connected MLP.

We use datasets across 56 cell types, comparing the above methods over each of the 56 different tasks.

**Model Hyperparameters:** For AttentiveChrome variations, we set the bin-level LSTM embedding size $d$ to 32 and the HM-level LSTM embedding size as 16. Since we implement a bi-directional LSTM, this results in each embedding vector $\mathbf{h}_t$ as size 64 and embedding vector $\mathbf{m}_j$ as size 32. Therefore, we set the context vectors, $\mathbf{W_b}$ and $\mathbf{W_s}$, to size 64 and 32 respectively.[7]

**Performance Evaluation:** Table 2 compares different variations of AttentiveChrome using summarized AUC scores across all 56 cell types on the test set. We find that overall the LSTM-attention based models perform better than CNN-based and LSTM baselines. CNN-attention model gives worst performance. To add the bin-level attention layer to the CNN model, we removed the max-pooling layer. We hypothesize that the absence of max-pooling is the cause behind its low performance. LSTM-$\alpha$ has better empirical performance than the LSTM-$\alpha, \beta$ model. We recommend the use of the proposed AttentiveChrome LSTM-$\alpha, \beta$ (from here on referred to as AttentiveChrome) for hypothesis generation because it provides a good trade-off between AUC and interpretability. Also, while the performance improvement over DeepChrome [29] is not large, AttentiveChrome is better as it allows interpretability to the "black box" neural networks.

Table 3: Pearson Correlation values between weights assigned for $H_{prom}$ (active HM) by different visualization techniques and $H_{active}$ read coverage (indicating actual activity near "ON" genes) for predicted "ON" genes across three major cell types.

| Viz. Methods | H1-hESC | GM12878 | K562 |
|---|---|---|---|
| $\alpha$ Map (LSTM-$\alpha$) | 0.8523 | **0.8827** | **0.9147** |
| $\alpha$ Map (LSTM-$\alpha, \beta$) | **0.8995** | 0.8456 | 0.9027 |
| Class-based Optimization (CNN) | 0.0562 | 0.1741 | 0.1116 |
| Saliency Map (CNN) | 0.1822 | -0.1421 | 0.2238 |

**Using Attention Scores for Interpretation:** Unlike images and text, the results for biology are hard to interpret by just looking at them. Therefore, we use additional evidence from REMC as well as introducing a new strategy to qualitatively and quantitatively evaluate the bin-level attention weights or $\alpha$-map LSTM-$\alpha$ model and AttentiveChrome. To specifically validate that the model is focusing its attention at the right bins, we use the read counts of a new HM signal - H3K27ac from REMC database. We represent this HM as $H_{active}$ because this HM marks the region that is active when the gene is "ON". H3K27ac is an important indication of activity in the DNA regions and is a good source to validate the results. We did not include H3K27ac Mark as input because it has not been profiled for all 56 cell types we used for prediction. However, the genome-wide reads of this HM mark are available for three important cell types in the blood lineage: H1-hESC (stem cell), GM12878 (blood cell), and K562 (leukemia cell). We, therefore, chose to compare and validate interpretation in these three cell types. This HM signal has not been used at any stage of the model training or testing. We use it solely to analyze the visualization results.

We use the average read counts of $H_{active}$ across all 100 bins and for all the active genes (gene=ON) in the three selected cell types to compare different visualization methods. We compare the attention $\alpha$-maps of the best performing LSTM-$\alpha$ and AttentiveChrome models with the other two popular visualization techniques: (1) the Class-based optimization method and (2) the Saliency map applied on the baseline DeepChrome-CNN model. We take the importance weights calculated by all visualization methods for our active input mark, $H_{prom}$, across 100 bins and then calculate their Pearson correlation to $H_{active}$ counts across the same 100 bins. $H_{active}$ counts indicate the actual active regions. Table 3 reports the correlation coefficients between $H_{prom}$ weights and read coverage of $H_{active}$. We observe that attention weights from our models consistently achieve the highest correlation with the actual active regions near the gene, indicating that this method can capture the important signals for predicting gene activity. Interestingly, we observe that the saliency map on the DeepChrome achieves a higher correlation with $H_{active}$ than the Class-based optimization method for two cell types: H1-hESC (stem cell) and K562 (leukemia cell).

Next, we obtain the attention weights learned by AttentionChrome, representing the important bins and HMs for each prediction of a particular gene as ON or OFF. For a specific gene sample, we can visualize and inspect the bin-level and HM-level attention vectors $\alpha_t^j$ and $\beta^j$ generated by AttentionChrome. In Figure 2(a), we plot the average bin-level attention weights for each HM for cell type GM12878 (blood cell) by averaging $\alpha$-maps of all predicted "ON" genes (top) and "OFF" genes (bottom). We see that on average for "ON" genes, the attention profiles near the TSS region are well defined for $H_{prom}$, $H_{enhc}$, and $H_{struct}$. On the contrary, the weights are low and close to uniform for $H_{reprA}$ and $H_{reprB}$. This average trend reverses for "OFF" genes in which $H_{reprA}$ and $H_{reprB}$ seem to gain more importance over $H_{prom}$, $H_{enhc}$, and $H_{struct}$. These observations make sense biologically as $H_{prom}$, $H_{enhc}$, and $H_{struct}$ are known to encourage gene activation while $H_{reprA}$ and $H_{reprB}$ are known to repress the genes [8]. On average, while $H_{prom}$ is concentrated near the TSS region, other HMs like $H_{struct}$ show a broader distribution away from the TSS. In summary, the importance of each HM and its position varies across different genes. E.g., $H_{enhc}$ can affect a gene from a distant position.

In Figure 2(b), we plot the average read coverage of $H_{active}$ (top) for the same 100 bins, that we used for input signals, across all the active genes (gene=ON) for GM12878 cell type. We also plot the bin-level attention weights $\alpha_t^j$ for AttentiveChrome (bottom) averaged over all genes predicted as ON for GM12878. Visually, we can tell that the average $H_{prom}$ profile is similar to $H_{active}$. This

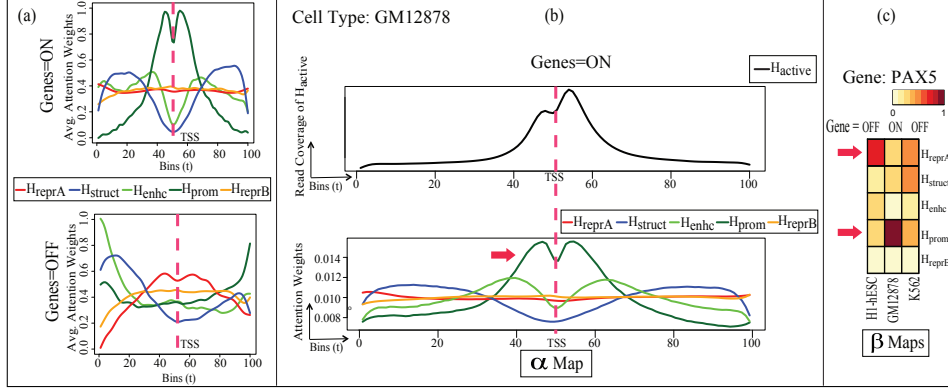

Figure 2: (Best viewed in color) (a) Bin-level attention weights ($\alpha_t^j$) from AttentiveChrome averaged for all genes when predicting gene=ON and gene=OFF in GM12878 cell type. (b) Top: Cumulative $H_{active}$ signal across all active genes. Bottom: Plot of the bin-level attention weights ($\alpha_t^j$). These weights are averaged for gene=ON predictions. $H_{prom}$ weights are concentrated near the TSS and corresponds well with the $H_{active}$ indicating actual activity near the gene. This indicates that AttentiveChrome is focusing on the correct bin positions for this case (c) Heatmaps visualizing the HM-level weights ($\beta^j$), with $j \in \{1, ..., 5\}$ for an important differentially regulated gene (PAX5) across three blood lineage cell types: H1-hESC (stem cell), GM12878 (blood cell), and K562 (leukemia cell). The trend of HM-level $\beta^j$ weights for PAX5 have been verified through biological literature.

observation makes sense because $H_{prom}$ is related to active regions for "ON" genes. Thus, validating our results from Table 3.

Finally in Figure 2(c) we demonstrate the advantage of AttentiveChrome over LSTM-$\alpha$ model by printing out the $\beta^j$ weights for genes with differential expressions across the three cell types. That is, we select genes with varying ON(+1)/OFF(−1) states across the three chosen cell types using a heatmap. Figure 2(c) visualizes the $\beta^j$ weights for a certain differentially regulated gene, PAX5. PAX5 is critical for the gene regulation when stem cells convert to blood cells ([25]). This gene is OFF in the H1-hESC cell stage (left column) but turns ON when the cell develops into GM12878 cell (middle column). The $\beta^j$ weight of repressor mark $H_{reprA}$ is high when gene=OFF in H1-hESC (left column). This same weight decreases when gene=ON in GM12878 (middle column). In contrast, the $\beta^j$ weight of the promoter mark $H_{prom}$ increases from H1-hESC (left column) to GM12878 (middle column). These trends have been observed in [25] showing that PAX5 relates to the conversion of chromatin states: from a repressive state ($H_{prom}$(H3K4me3):−, $H_{reprA}$(H3K27me3):+) to an active state ($H_{prom}$(H3K4me3):+, $H_{reprA}$(H3K27me3):−). This example shows that our $\beta^j$ weights visualize how different HMs work together to influence a gene's state (ON/OFF). We would like to emphasize that the attention weights on both bin-level ($\alpha$-map) and HM-level ($\beta$-map) are gene (i.e. sample) specific.

The proposed AttentiveChrome model provides an opportunity for a plethora of downstream analyses that can help us understand the epigenomic mechanisms better. Besides, relevant datasets are big and noisy. A predictive model that automatically selects and visualizes essential features can significantly reduce the potential manual costs.

## 6  Conclusion

We have presented AttentiveChrome, an attention-based deep-learning approach that handles prediction and understanding in one architecture. The advantages of this work include:

- AttentiveChrome provides more accurate predictions than state-of-the-art baselines (Table 2).
- The attention scores of AttentiveChrome provide a better interpretation than saliency map and class optimization (Table 3). This allows us to view what the model 'sees' when making its prediction.
- AttentiveChrome can model highly modular feature inputs in which each is sequentially structured.
- To the authors' best knowledge, AttentiveChrome is the first implementation of deep attention mechanism for understanding data about gene regulation. We can gain insights and understand the predictions by locating 'what' and 'where' AttentiveChrome has focused (Figure 2). Many real-world applications are seeking such knowledge from data.

## Footnotes

[1] Code shared at www.deepchrome.org.

[2] DNA is a long string of paired chemical molecules or nucleotides that fall into four different types and are denoted as A, T, C, and G. DNA carries information organized into units such as genes. The set of genetic material of DNA in a cell is called its genome.

[3] The complex of DNA, histones, and other structural proteins is called chromatin.

[4] In biology this field is called epigenetics. "Epi" in Greek means over. The epigenome in a cell is the set of chemical modifications over the chromatin that alter gene expression.

[5] A base pair refers to one of the double-stranded DNA sequence characters (ACGT)

[6]We tried several different architectures to model the dependencies among HMs, and found no clear ordering.

[7]We can view $\mathbf{W_b}$ as $1 \times 64$ matrix.

[8]The small dips at the TSS in both subfigures of Figure 2(a) are caused by missing signals at the TSS due to the inherent nature of the sequencing experiments.

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
