[Supplementary Material]

# Attend and Predict: Understanding Gene Regulation by Selective Attention on Chromatin

**Ritambhara Singh, Jack Lanchantin, Arshdeep Sekhon, Yanjun Qi**
Department of Computer Science
University of Virginia
`yanjun@virginia.edu`

## S:1  Supplementary Information

### S:1.1  More about results

**Importance of All HMs as input signals:**  Not all HMs carry the same information, and it is important to include different HMs for gene expression prediction. While $H_{prom}$ may be essential to predict gene=ON, $H_{enhc}$ may play a role to make that prediction. Contrarily, for "OFF" genes, HMs like $H_{reprA}$ may play a significant role. To demonstrate this, we used only one HM at a time and performed the classification. The accuracy decreases when just one HM is used. Table S:1 shows AUC scores in GM12878 when all HMs are used as input signals and when we use them one at a time. We observe that the performance drops drastically, indicating that it is vital to include different HMs for gene expression prediction.

Table S:1: AUC scores in GM12878 when all HMs are used as input signals and when we use them one at a time. The AUC score reduces drastically, indicating that it is vital to include different HMs for gene expression prediction

| HMs used as input | AUC Score |
|---|---|
| All 5 HMs | 0.9085 |
| $H_{prom}$ | 0.8893 |
| $H_{enhc}$ | 0.8516 |
| $H_{struct}$ | 0.8506 |
| $H_{reprA}$ | 0.7698 |
| $H_{reprB}$ | 0.6465 |

### S:1.2  More about experimental settings

**Evaluation Metric for Classification:**  We use the area under the receiver operating characteristic curve (AUC) as our evaluation metric. AUC represents the probability that a randomly selected 'event' will be regarded with greater suspicion than a randomly selected 'non-event'. AUC scores range between 0 and 1, with values closer to 1 indicating successful predictions.

**Choice for Evaluation Metric:**  We also calculated the F1-scores for baseline DeepChrome[2] and AttentiveChrome, presented in Table S:2.

We observe that the F1-scores vary significantly across cell types. This is because, for most cell types, the number of samples with Gene=OFF are much more substantial (80% of Test data) than those with Gene=ON. Since AUC score, used in all previous works, is independent of the distribution of positive and negative classes and depends only on the learning model, we use it as our evaluation metric to measure performances of different models.

Table S:2: Performance comparison in terms of F1-scores

|  | DeepChrome[2] | AttChrome |
|---|---|---|
| Mean | 0.55 | **0.56** |
| Median | **0.69** | 0.62 |
| Max | **0.89** | 0.88 |
| Min | 0.12 | **0.16** |

**Choice of Baselines:** We chose the DeepChrome CNN based model [2] as our baseline, as it has been shown to outperform SVM and Random Forest based models used previously for this task (reported in [2]) . We summarize performance results of all the baselines in Table S:3.

Table S:3: Performance comparison (in AUC Scores) of all baseline models

|  | RF | SVM | DeepChrome[2] | LSTM |
|---|---|---|---|---|
| Mean | 0.59 | 0.75 | 0.80 | **0.81** |
| Median | 0.58 | 0.76 | **0.80** | **0.80** |
| Max | 0.71 | 0.87 | **0.92** | 0.92 |
| Min | 0.52 | 0.62 | 0.69 | **0.71** |

### S:1.3  More about Method

**Long Short-Term Memory (LSTM) Networks:** Recurrent neural networks (RNNs) have been designed for modeling sequential data samples and are used widely in sequential data application tasks such as natural language processing. RNNs are advantageous over CNNs because they can capture the complete set of dependencies among spatial positions in a sequential sample.

Given an input matrix $\mathbf{X}$ of size $n_{in} \times T$, an RNN produces a matrix $\mathbf{H}$ of size $d \times T$, where $n_{in}$ is the input feature size, $T$ is the input feature length, and $d$ is the RNN embedding size. At each timestep $t \in [1..T]$, an RNN takes an input column vector $\mathbf{x_t} \in \mathbb{R}^{n_{in}}$ and the previous hidden state vector $\mathbf{h_{t-1}} \in \mathbb{R}^d$ and produces the next hidden state $\mathbf{h_t}$ by applying the following recursive operation:

$$\mathbf{h}_t = \sigma(\mathbf{W}x_t + \mathbf{U}\mathbf{h}_{t-1} + \mathbf{b}), \tag{S:1-1}$$

where $\mathbf{W}, \mathbf{U}, \mathbf{b}$ are the trainable parameters of the model, and $\sigma$ is an element-wise nonlinearity function. Due to their recursive nature, RNNs can model the full conditional distribution of any sequential data and find dependencies over time. To handle "vanishing gradient" issue of training basic RNNs, [1] proposed an RNN variant called the Long Short-term Memory (LSTM) network,which can handle long term dependencies by using gating functions. These gates can control when information

Figure S:1: A simple representation of an LSTM module.

is written to, read from, and forgotten. Specifically, LSTM "cells" take inputs $\mathbf{x}_t, \mathbf{h}_{t-1}$, and $\mathbf{c}_{t-1}$, and produce $\mathbf{h}_t$, and $\mathbf{c}_t$:

$$\mathbf{i}_t = \sigma(\mathbf{W}^i\mathbf{x}_t + \mathbf{U}^i\mathbf{h}_{t-1} + \mathbf{b}^i)$$
$$\mathbf{f}_t = \sigma(\mathbf{W}^f\mathbf{x}_t + \mathbf{U}^f\mathbf{h}_{t-1} + \mathbf{b}^f)$$
$$\mathbf{o}_t = \sigma(\mathbf{W}^o\mathbf{x}_t + \mathbf{U}^o\mathbf{h}_{t-1} + \mathbf{b}^o)$$
$$\mathbf{g}_t = tanh(\mathbf{W}^g\mathbf{x}_t + \mathbf{U}^g\mathbf{h}_{t-1} + \mathbf{b}^g)$$
$$\mathbf{c}_t = \mathbf{f}_t \odot \mathbf{c}_{t-1} + \mathbf{i}_t \odot \mathbf{g}_t$$
$$\mathbf{h}_t = \mathbf{o}_t \odot tanh(\mathbf{c}_t)$$

where $\sigma(\cdot), tanh(\cdot)$, and $\odot$ are element-wise sigmoid, hyperbolic tangent, and multiplication functions, respectively. $\mathbf{i}_t, \mathbf{f}_t$, and $\mathbf{o}_t$ are the input, forget, and output gates, respectively.

---

**Algorithm S:1** AttentiveChrome: Forward Propagation

---
**Require:** $X$                                                                    ▷ Size: $M \times T$

---
1: **procedure** CLASSIFICATION($X$)
2:     $\{x_1^t, x_2^t, \ldots x_j^t\} \leftarrow X$                       ▷ Size: $1 \times T$, $t \in \{1, \ldots T\}$ and $j \in \{1, \ldots M\}$
3:     $\mathbf{m}^j \leftarrow BinLevelAttention(x_t^j)$
4:     $\mathbf{v} \leftarrow HMLevelAttention(\mathbf{m}^j)$
5:     $y \leftarrow MultiLayerPerceptron(\mathbf{v})$
6: **return** $y$
7: **procedure** BIN-LEVEL ATTENTION($x_t^j$)
8:     **for** $j \in \{1, \ldots M\}$ **do**                                    ▷ Run in Parallel
9:
10:        $\overrightarrow{\mathbf{h}_t^j} \leftarrow \overrightarrow{LSTM}^j(x_t^j)$                             ▷ Bi-directional LSTM
11:        $\overleftarrow{\mathbf{h}_t^j} \leftarrow \overleftarrow{LSTM}^j(x_t^j)$
12:        $\mathbf{h}_t^j \leftarrow [\overrightarrow{\mathbf{h}_t^j}, \overleftarrow{\mathbf{h}_t^j}]$.
13:        $\alpha_t^j \leftarrow \frac{exp(\mathbf{W}_b \mathbf{h}_t^j)}{\sum_{i=1}^{T} exp(\mathbf{W}_b \mathbf{h}_i^j)}$         ▷ Size: $1 \times T$ for each $j \in \{1, \ldots M\}$
14:        $\mathbf{m}^j \leftarrow \sum_{t=1}^{T} \alpha_t^j \times \mathbf{h}_t^j$
        **return** $\mathbf{m}^j$
15: **procedure** HM-LEVEL ATTENTION($\mathbf{m}^j$)
16:        $\mathbf{s}^j \leftarrow [\overrightarrow{LSTM_s}(\mathbf{m}^j), \overleftarrow{LSTM_s}(\mathbf{m}^j)]$
17:        $\beta^j \leftarrow \frac{exp(\mathbf{W}_s \mathbf{s}^j)}{\sum_{i=1}^{M} exp(\mathbf{W}_s \mathbf{s}^i)}$                    ▷ Size: $1 \times M$
18:        $\mathbf{v} \leftarrow \sum_{j=1}^{M} \beta^j \mathbf{s}^j$
        **return** $\mathbf{v}$

---

**AttentiveChrome Details:** AttentiveChrome Forward Propagation algorithm is presented in Algorithm box S:1, while Figure S:2 presents the overview of the proposed AttentiveChrome in detail.

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

Figure S:2: Overview of the proposed AttentiveChrome, a unified framework that can both predict and understand how histone modifications regulate gene expression. We present six steps in order: (a) We generate an input matrix $\mathbf{X}$ for each gene's TSS flanking region, consisting of 100 bins as rows and 5 histone modification (HM) signals as columns. (b) We split the matrix into five vectors representing each HM mark. We input these vectors into the AttentiveChrome model. (c) We use a separate LSTM to learn feature representations of an HM mark. (d) A bin-level attention layer is learned to extract bins that are important for representing an HM mark. This attention layer will aggregate important bins to form an embedding vector for an HM. Here we only show the case of HM2 in steps (c) and (d). (e) Next, to capture the dependencies among different HM marks, we apply another LSTM layer over the representation of 5 HMs. (f) To reward HM marks that are significant clues for classifying an individual gene's expression, AttentiveChrome adds another attention layer- HM-level attention. This layer outputs an embedding vector $\mathbf{v}$ for the whole gene region under consideration. (g) Finally, the output embedding $\mathbf{v}$ from the previous layers will be fed into a classification module to predict the gene expression as high(+1)/low(-1).