[Reviews · NeurIPS 2017]

Reviewer 1



Aim The work aim was to classify gene expression over 56 different type of cells based on histones and histones modification marks considered on the whole genome. They used an attention-based deep learning with a Long Short-Term Memory (LSTM) module to model the dependencies. The attention-base deep learning is needed to enlight the more significant features in the input matrix, while LSTM is used for the encoding. Comments The proposed method provides the possibility to interpret the results and to detect significant features over all the genome considering also the position. The method works better than previously proposed method in terms of classification score, even though the authors only use the accuracy as performance score. It may be worth commenting also on other metrics. In section 3 a direct reference to the figures in supplementary materials would have allowed a better understanding. It would be interesting to see the behaviour in case of regression since setting the expression as ON and OFF is an over-semplification of the problem.

Reviewer 2



The paper presents a novel method for predicting gene regulation by LSTM with an attention mechanism. The model consists of two levels, where the first level is applied on bins for each histone modifications (HM) and the second level is applied to multiple HMs. Attention mechanism is used in each level to focus on the important parts of the bins and HMs. In the experiments, the proposed method improves AUC scores over baseline models including CNN, LSTM, and CNN with an attention mechanism. This is an interesting paper which shows that LSTM with an attention mechanism can predict gene regulation. 1. It is unclear to me why the second level is modeled using LSTM because there is no ordering between HMs. Would it be reasonable to use a fully connected layer to model dependencies between HMs? Based on Table 1 and 2, the one level model (LSTM-\alpha) outperforms the two level model ((LSTM-\alpha,\beta) in many cases. It would be interesting to investigate how HMs are coupled with each other in the learned model. 2. In Section 2, an algorithm box that includes the equations for the entire model would be very helpful. 3. It is unclear to me if DeepChrome was compared with the proposed method in the experiments. It would be helpful to indicate which model corresponds to DeepChrome. 4. As baselines, it would be helpful to include non-neural network models such as SVM or logistic regression. 5. To allow researchers in biology to use this method, it would be very helpful to have the source code publicly available with URLs in the paper.